# Transcriptome Analysis and Functional Characterization of the *HvLRR_8-1* Gene Involved in Barley Resistance to *Pyrenophora graminea*

**DOI:** 10.3390/plants14152350

**Published:** 2025-07-30

**Authors:** Wenjuan Yang, Ming Guo, Yan Li, Qinglan Yang, Huaizhi Zhang, Chengdao Li, Juncheng Wang, Yaxiong Meng, Xiaole Ma, Baochun Li, Lirong Yao, Hong Zhang, Ke Yang, Xunwu Shang, Erjing Si, Huajun Wang

**Affiliations:** 1State Key Laboratory of Aridland Crop Science, Gansu Key Laboratory of Crop Improvement and Germplasm Enhancement, Lanzhou 730070, China; yangwj7227@163.com (W.Y.); 18794807552@163.com (M.G.); 17393376279@163.com (Y.L.); 13613771310@163.com (Q.Y.); zhzhyf0802@163.com (H.Z.); wangjc@gsau.edu.cn (J.W.); yxmeng1@163.com (Y.M.); maxl@gsau.edu.cn (X.M.); libc@gsau.edu.cn (B.L.); ylr0384@163.com (L.Y.); zhanghong@gsau.edu.cn (H.Z.); yk_831116@163.com (K.Y.); 17393294106@163.com (X.S.); 2College of Agronomy, Gansu Agricultural University, Lanzhou 730070, China; 3Western Barley Genetics Alliance, College of Science, Health, Engineering and Education, Murdoch University, Murdoch, WA 6150, Australia; c.li@murdoch.edu.au; 4College of Life Science and Technology, Gansu Agricultural University, Lanzhou 730070, China

**Keywords:** *Hordeum vulgare* L., barley leaf stripe, *Pyrenophora graminea*, transcriptome, *HvLRR_8-1*, BSMV-VIGS

## Abstract

Barley leaf stripe, caused by *Pyrenophora graminea* (*Pg*), significantly reduces yields across various regions globally. Understanding the resistance mechanisms of barley to *Pg* is crucial for advancing disease resistance breeding efforts. In this study, two barley genotypes—highly susceptible Alexis and immune Ganpi2—were inoculated with the highly pathogenic *Pg* isolate QWC for 7, 14, and 18 days. The number of differentially expressed genes (DEGs) in Alexis was 1350, 1898, and 2055 at 7, 14, and 18 days, respectively, while Ganpi2 exhibited 1195, 1682, and 2225 DEGs at the same time points. Gene expression pattern analysis revealed that Alexis responded more slowly to *Pg* infection compared to Ganpi2. A comparative analysis identified 457 DEGs associated with Ganpi2’s immunity to *Pg*. Functional enrichment of these DEGs highlighted the involvement of genes related to plant-pathogen interactions and kinase activity in *Pg* immunity. Additionally, 20 resistance genes and 24 transcription factor genes were predicted from the 457 DEGs. Twelve candidate genes were selected for qRT-PCR verification, and the results showed that the transcriptomic data was reliable. We conducted cloning of the candidate *Pg* resistance gene *HvLRR_8-1* by the barley cultivar Ganpi2, and the sequence analysis confirmed that the *HvLRR_8-1* gene contains seven leucine-rich repeat (LRR) domains and an S_TKc domain. Subcellular localization in tobacco indicates that the HvLRR_8-1 is localized on the cell membrane. Through the functional analysis using virus-induced gene silencing, it was demonstrated that *HvLRR_8-1* plays a critical role in regulating barley resistance to *Pg*. This study represents the first comparative transcriptome analysis of barley varieties with differing responses to *Pg* infection, providing that *HvLRR_8-1* represents a promising candidate gene for improving durable resistance against *Pg* in cultivated barley.

## 1. Introduction

Barley leaf stripe (BLS), caused by *Pyrenophora graminea* (Ito and Kuribayashi) [anamorph *Drechslera graminea* (Rabenh. ex. Schlech.) Shoemaker] (*Pg*), is a seed-borne disease of barley that is destructive in most barley-growing regions worldwide. This disease has been reported in many countries and regions, including Europe, North America, North Africa, Russia, India, and China [1]. The mycelia of the pathogen can survive in the pericarp, hull, and seed coat but not in the embryo. When the seed germinates, the hyphae grow in the intercellular space of the coleorhizae, initiating a systemic infection by penetrating the root tip [2,3]. The toxins produced by *Pg* lead to the formation of longitudinal dark-brown necrotic stripes between the leaf veins and can also cause spike sterility [4,5,6].

BLS can result in significant yield losses, especially when the disease incidence is high [7,8]. Chemical treatments can prevent BLS attacks to some extent [9], but achieving an acceptable yield is unlikely if more than 30% of seeds are infected, unless the barley cultivar possesses strong resistance [10]. Additionally, chemical treatments may reduce the number of emergent seedlings and negatively impact the environment [11], making the use of host resistance a more effective and sustainable approach.

Although breeding for durable resistance is an ideal strategy for disease control, it requires resistant materials or resistance genes, as well as an understanding of the interactions between barley and *Pg*. Barley exhibits a wide range of resistance to *Pg*, and resistant lines have been identified through both artificial inoculation and natural infection [12,13]. To date, only three resistance genes have been mapped [14,15,16], and several quantitative trait loci (QTLs) associated with resistance have been identified [12,17,18,19,20]. However, little is known about the specific mechanisms underlying resistance. For example, the *Rdg2a* gene has been shown to inhibit the growth of the virulent *Pg* isolate Dg2 on the embryo-distal side of the scutellum [3]. Additionally, another study using differential display reverse-transcription polymerase chain reaction (DDRT-PCR) identified 15 up-regulated genes involved in the immune response of a *Pg*-resistant genotype, with callose deposition playing a key role in resistance against *Pg* [21]. *Pg* also exhibits genetic diversity, as evidenced by different molecular marker types [22,23,24] and pathogenicity variations observed through artificial inoculation [13,23,24]. Despite ongoing research, the precise mechanism responsible for the salt tolerance of cotton remains a mystery. Despite ongoing research, several resistance genes against BLS disease have been mapped and cloned; the mechanisms by which these genes activate barley’s disease resistance responses remain unclear [25].

With the advancement of RNA-Seq, it has been widely used to identify resistance factors in various plants infected by pathogens. and has become an essential tool for deciphering plant-pathogen interactions [26]. Similarly, a substantial repertoire of genes controlling critical disease-resistance traits in barley has been characterized through RNA-Seq analyses. Zeng et al. [27] identified 831 powdery mildew-responsive differentially expressed genes in Tibetan barley using RNA-Seq. Wang et al. [28] identified four pairs associated with barley leaf stripe miRNAs and target genes through RNA-Seq, combined with miRNA-seq and degradation-seq: The differential expression of *Hvu-miR168-5p* and *HvAGO1*, *Hvu-novel-52* and *HvGRF6*, *Hvu-miR6195* and *CLP*, and *Hvu-miR159b* and *GAMYB* triggered the expression of key target genes. Similarly, transcriptome analysis of barley leaf spot, caused by *Ramularia collocygni*, demonstrated the critical role of xylanase inhibitors during early pathogen recognition [29]. In recent years, transcriptomic sequencing has also been widely used to analyze changes in gene expression related to the barley-pathogen interaction and has also proven to be pivotal in identifying core pathways associated with disease resistance in barley. Mehtap et al. [30] integrated transcriptomic profiling in the *Bipolaris sorokiniana*-resistant barley cultivar Dara revealed significant upregulation of the *PR3* gene, highlighting the role of chitinase-mediated cell wall reinforcement in defense. This resistance mechanism functions through degradation of the invading pathogen’s cell wall components. Additionally, Ghannam et al. [21] employed RNA-Seq to identify 15 barley key disease-resistant genes potentially involved in *Pg*-immune responses, while also revealing that oxidative metabolic processes such as reactive oxygen species (ROS)-related pathways may contribute to disease resistance mechanisms. In barley resistant to powdery mildew *Blumeria graminis*, *PR1* and *PR5* in the salicylic acid (SA) pathway were significantly upregulated, while the jasmonic acid (JA) pathway appeared to be suppressed by pathogen effector proteins [31]. Another study demonstrated that in ROS metabolism, spot blotch-resistant cultivars rapidly activated the ROS scavenging system to restrict pathogen proliferation [32]. Plants utilize various types of resistance R-genes to detect the presence of pathogens and induce defense responses against invasions by fungal, bacterial, and other pathogenic organisms [33]. To date, a substantial number of R-genes have been identified across diverse plant species. The largest known class of R-genes encodes proteins containing nucleotide-binding site (NBS) and leucine-rich repeat (LRR) domains [34]. Upon perception of external stress signals by these R-genes, the activity of leucine-rich repeat (LRR) domains is significantly enhanced. Subsequently, these activated proteins interact with receptor-like kinases (RLKs) and other receptors to form functional kinase complexes [35]. As the most abundant class of plant receptors, LRR-RLKs play pivotal roles in signal transduction and directly regulate various developmental processes. Among these, leucine-rich repeat receptor-like kinases (LRR-RLKs) represent a central component of plant pattern recognition receptors (PRRs) [36,37]. These receptors utilize their extracellular LRR domains to recognize pathogen-associated molecular patterns (PAMPs) or effector proteins, while their intracellular serine/threonine kinase (S_TKc) domains mediate the activation of downstream defense signaling cascades [38]. As the LRR domain interacts with pathogenic effectors, it likely determines resistance specificity, while putative solvent-exposed residues in β-sheets may interact with pathogen ligands, thereby determining the specificity of pathogen elicitors [39]. Researchers have found that the leucine-rich repeat (LRR) domain typically comprises 20–29 amino acid residues and contains 11 consensus sequences, LxxLxLxxN/CxL, in which any amino acid is “x”; “L” is Val, Leu, or Ile; “N” is Asn, Thr, Ser, or Cys; and “C” is Cys or Ser [40,41]. Recent studies have demonstrated that certain LRR proteins containing relatively few LRR repeat units 5–10 repeats can still effectively initiate immune responses through interactions with co-receptors or formation of hetero-oligomeric complexes [42]. Meanwhile, several studies have demonstrated that these LRR receptor-like genes play crucial roles in plant adaptation and function across various physiological processes, including development, growth, and responses to both abiotic and biotic stresses [43]. Consequently, LRR proteins can activate protective immune signaling upon detection of potential pathogens, thereby providing an early warning system in plants [44].

Since Holzberg et al. [45] first successfully utilized Barley stripe mosaic virus (BSMV) as a vector to suppress Phytoene desaturase (*PDS*) gene expression in monocot barley, this groundbreaking work has not only established a novel approach for functional gene studies in barley but also laid the foundation for the application of virus-induced gene silencing (VIGS) technology in other monocotyledonous plants [46]. Subsequently, VIGS technology enables functional genomic studies in plants at the genetic level [47]. The successful establishment of BSMV-mediated VIGS in barley marked a significant advancement, particularly for cereal crops where conventional transgenic approaches face limitations due to technical challenges, cultivar specificity, and substantial time/labor requirements. This technology has become indispensable for disease resistance research, facilitating functional characterization of key wheat resistance genes, including leaf rust resistance genes *(Lr21/Lr10*) [48] and stem rust resistance genes (*TaRLK1/2*) [49] in wheat, as well as the powdery mildew resistance gene (*Hsp90*) in barley [50].

In this study, we performed comparative transcriptome analysis at different time points using resistant and susceptible barley materials exhibiting differential responses to *Pg* infection. We identified and cloned a leucine-rich repeat receptor-like kinase (LRR-RLK) gene, *HvLRR_8-1*, from barley, which encodes a protein containing seven leucine-rich repeat (LRR) domains and one serine/threonine kinase catalytic (S_TKc) domain. Subsequently, we employed a virus-induced gene silencing (VIGS) system to evaluate its role in *Pg* resistance, with the results further validated by trypan blue staining and 3, 3′ -diaminobenzidine (DAB) staining. This research aims to identify the resistance factors in barley against *Pg*, which could provide new targets for effective disease control strategies. We present for the first time the transcriptome response of immune and highly susceptible barley varieties to *Pg*, identifying a leucine-rich-repeat receptor-like kinase (LRR-RLKs) candidate gene associated with resistance to *Pg*.

## 2. Result

### 2.1. Microscopic Observation of Symptoms and Resistant Spectrum Detection of Ganpi2 to Pg Isolates

In the field, the typical stripe symptom was observed (Figure 1A), and BLS could cause severely diseased plants to be unable to bring out their ears (Figure 1B). The conidium of *Pyrenophora graminea* scraped from the site of the yellow stripe was observed (Figure 1C). To further observe the site of the yellow stripe, mycelium was observed in different magnifications (Figure 1D–G). The *Pg* isolate QWC could result in a high incidence in Alexis and cause typical stripe symptoms (Figure 1H). However, it could not infect Ganpi2, so Ganpi2 is immune to *Pg* isolate QWC.

To confirm the resistance of Ganpi2 to other *Pg* isolates from China, Ganpi2 was inoculated with 28 *Pg* isolates using the “sandwich” method [19]. The results showed that Ganpi2 exhibited immunity to most isolates, except for isolate BS-13001-2, which had an incidence rate of 2.56% (Table 1). This finding indicates that Ganpi2 possesses broad-spectrum resistance to *Pg* isolates from different regions of China.

### 2.2. RNA-Seq Data, Assessment of BGIseq500 Sequence, and Aligning to Reference Genomes

To investigate the gene expression induced in the two barley varieties, a total of 36 libraries were constructed and sequenced. These libraries included immune Ganpi2 and highly susceptible Alexis, inoculated for 7, 14, and 18 days, along with their mock-inoculated controls at the corresponding time points. The sequencing was performed on three biological replicates of infected and corresponding healthy embryos at each stage. All sequence data have been deposited in the NCBI-SRA (https://www.ncbi.nlm.nih.gov/sra, accessed on 15 April 2021) database and are accessible under accession numbers SRA14555703-14555720 for Alexis and SRA14514353-14514370 for Ganpi2. The average size of clean bases for each sample was 10.52 Gb. A total of 2881.66 million raw reads were generated from the 36 libraries. After filtering out reads with adapters, reads with unknown base content (N) greater than 5%, and low-quality reads, 2524.71 million clean reads were obtained, with Q20 values greater than 95.29. The average clean read ratio for all samples was 87.62%. The clean reads were aligned to the reference genome, including barley genomes, using HISAT 2015 software [51], with a mapping ratio for each library ranging from 85.12% to 87.06% and an average of 86.30% (Appendix A). Bowtie2 [52] was used to align clean reads to reference gene sequences, resulting in a total mapping ratio ranging from 68.94% to 72.85% (Appendix A).

### 2.3. Expression Pattern Analysis and Functional Enrichment of DEGs

The DEG profiles were obtained for two genotypes, including *Pg*-infected and control samples, in response to *Pg* infection. In Alexis, comparing *Pg*-infected embryos with control embryos at T7, T14, and T18 revealed 801, 1032, and 1589 up-regulated DEGs and 549, 866, and 466 down-regulated DEGs, respectively. In Ganpi2, 867, 1191, and 1118 up-regulated DEGs were detected at T7, T14, and T18, along with 328, 491, and 1107 down-regulated DEGs, respectively (Figure 2A, Appendix A).

To investigate the differences in DEG expression patterns between Alexis and Ganpi2, Short Time-series Expression Miner (STEM) v 1.3.12 software was used to analyze expression trends at different time points. The gene expression trends in response to *Pg* infection differed between Alexis and Ganpi2. In Alexis, the peak of gene expression occurred at T18, while in Ganpi2, it was at T14. Thus, T14 represents a significant time point for changes in gene expression levels in Ganpi2 (profiles 6, 5, and 1; *p* < 0.05, Figure 2B), whereas significant changes in Alexis were observed at T18 (profiles 7, 3, and 4; *p* < 0.05, Figure 2B). This suggests that Ganpi2 has a faster transcriptional response to *Pg* infection than Alexis. Given the substantial changes in gene expression in Ganpi2 at T14, this time point may represent a critical period for the resistance response to *Pg* (Appendix A).

Enrichment analysis was conducted to display the functional differences in gene expression changes associated with overrepresented profiles between the two genotypes. In Ganpi2, genes with consistently high expression levels at T14 and T18 were mainly involved in catalytic activity, ion binding, phenylpropanoid biosynthesis, and plant-pathogen interaction (profile 6, Figure 2B). Genes peaking at T14 were primarily enriched in oxidoreductase activity, starch and sucrose metabolism, and photosynthesis (profile 5, Figure 2B). In Alexis, genes were enriched in catalytic activity, response to stimuli, phenylpropanoid biosynthesis, and plant-pathogen interaction, with a continual increase in expression (profile 7, Figure 2B). In profile 3, genes were mainly involved in biological regulation, RNA transport, and the mRNA surveillance pathway; gene expression levels were not significantly up-regulated or down-regulated at T14 but were significantly down-regulated at T18. These results indicate differences in expression patterns and enrichment analyses, including GO and KEGG, between the two genotypes (Appendix A).

### 2.4. Detection of Candidate Genes Related to Ganpi2 Immune Response

#### 2.4.1. DEGs Might Be Related to Resistance to *Pg*

In Alexis, there were 2399 DEGs identified between A_7d_T and A_14d_T and 1898 DEGs between A_14d_CK and A_14d_T, with 583 DEGs common between these two comparisons. In Ganpi2, 5617 DEGs were detected between G_7d_T and G_14d_T, and 1682 DEGs between G_14d_CK and G_14d_T, with 1270 DEGs shared between these comparisons. Notably, 700 DEGs were found exclusively in Ganpi2 that were common between G_7d_T-G_14d_T and G_14d_CK-G_14d_T but absent in both A_7d_T-A_14d_T and A_14d_CK-A_14d_T in Alexis (Figure 2C, Appendix A). When comparing DEGs between Alexis and Ganpi2, 4357 DEGs were identified between A_14d_T and G_14d_T, and 2538 DEGs between A_14d_CK and G_14d_CK. Among these, 2716 DEGs were specific to the comparison between A_14d_T and G_14d_T. Of the 700 DEGs specific to Ganpi2, 457 DEGs were also found among the 2716 DEGs unique to the A_14d_T and G_14d_T comparison (Figure 2D, Appendix A). These DEGs may be associated with the resistance observed in Ganpi2.

#### 2.4.2. DEGs Involved in Plant-Pathogen Interactions

To further elucidate the functions of the 457 DEGs potentially associated with resistance in Ganpi2, KEGG enrichment analysis was performed. The analysis revealed that these DEGs were enriched in 93 pathways, with 11 pathways showing significant enrichment, including those related to resistance and plant-pathogen interactions (Figure 3A; Appendix A). To better understand the biological pathways involved in response to *Pg* infection, DEGs associated with resistance-related pathways were further examined.

During *Pg* infection, 27 DEGs were mapped to the plant-pathogen interaction pathway (Appendix A; Figure 3C). Additionally, genes encoding two pathogenesis-related protein 1 and six disease resistance proteins (RPM1 and RPS2) were detected, with peak expression at 18 days in Alexis and 14 days in Ganpi2. Of the 27 genes involved in plant-pathogen interaction, eight encoding receptor-like kinases (RLKs) were identified: one encoding LRR receptor-like serine/threonine-protein kinase EFR, six encoding LRR receptor-like serine/threonine-protein kinase FLS2, and one encoding receptor kinase-like protein XA21. The expression levels of these RLK-encoding genes peaked at 18 days in Alexis and 14 days in Ganpi2, except for *HORVU2Hr1G118010*, which peaked at 18 days in Ganpi2. Three transcription factors (TFs) were also identified, including two members of the WRKY family: WRKY TF 2/33 (*HORVU1Hr1G027700*) and WRKY TF 1/3/33 (*HORVU1Hr1G080300* and *HORVU7Hr1G113850*).

#### 2.4.3. DEGs Involved in Kinase Activation

GO enrichment analysis of the 457 DEGs potentially associated with resistance in Ganpi2 revealed enrichment in 855 GO terms, with 11 terms significantly enriched (Figure 3B; Appendix A). The significantly enriched terms mainly included kinase-related activities, such as protein kinase activity, kinase activity, and protein serine/threonine kinase activity. These results suggest that kinases may play a crucial role in resistance to *Pg*.

#### 2.4.4. DEGs Related to Resistance and TF

A total of 3108 putative resistance genes were predicted using DIAMOND [53], as detailed in Appendix A. Most of these resistance genes belonged to the RLP class, followed by the NL class. Given the important role of transcription factors (TFs) in regulating plant resistance under abiotic stress [54], 2025 TFs were predicted (Appendix A), with the MYB family being the most abundant, followed by the AP2-EREBP family.

To identify candidate resistance genes responsive to *Pg* invasion, 82 of the 457 DEGs were annotated as resistance genes, primarily containing domains from the NL, N, and RLP classes. Of the 111 putative resistance genes, 13 were derived from barley: six RLK-GNK2, three categorized as “other,” two from the NL class, one from the N class, and one RLP (Appendix A). The predominance of RLK-GNK2 suggests that RLKs may play a key role in the response to *Pg*.

Transcription factors play significant roles in regulating plant responses to abiotic stress. Among the 457 DEGs, 24 genes were identified as putative TFs. Most of these genes were assigned to the WRKY, NAC, and AP2-EREBP TF families, indicating their potential involvement in disease response (Appendix A).

### 2.5. qRT-PCR Verification of Differentially Expressed Genes

To verify the reliability of RNA-Seq results, the twelve key resistance genes in the pathway of plant-pathogen interaction were picked to verify the expression levels by qRT-PCR. RNA-Seq analysis results indicated that these genes were differentially upregulated in the immune cultivar Ganpi2 as compared to the susceptible cultivar Alexis. Barley embryos uninoculated with *Pg* strain served as controls. Across T7, T14, and T18 three stages, the results showed that on the 14th day of *Pg* infection, the expression of *HORVU7Hr1G115410*, *HORVU7Hr1G033620*, *HORVU7Hr1G075960*, and *HORVU5Hr1G066260* peaked in the immune cultivar Ganpi2, significantly higher than in the susceptible cultivar Alexis. The relative expression of *HORVU7Hr1G113850*, *HORVU4Hr1G086470*, *HORVU4Hr1G001060*, *HORVU6Hr1G010730*, and *HORVU2Hr1G118010* in immune cultivar Ganpi2 increased significantly with the progression of *Pg* infection, peaking on the 18th day. The expression trends of the twelve genes observed in RNA-Seq and qRT-PCR analyses were consistent (Figure 3D). These results indicate that the transcriptional data accurately reflects the response of barley to *Pg*.

### 2.6. Cloning and Characterization of HvLRR_8-1 Gene

Integrated analyses of transcription profiling, functional annotation, and disease resistance relevance revealed that receptor-like kinases (RLKs) play pivotal roles in plant disease resistance [43,44]. It is essential to identify and functionally characterize the corresponding genes from the resistance to *Pg* in Ganpi2. The *HvLRR_8-1* (*HORVU2Hr1G118010*) gene, annotated as receptor-like kinase, was selected for the next functional analysis. As a result, a 3474 bp band was successfully cloned by PCR-specific amplification. The results showed that the *HvLRR_8-1* gene was successfully cloned from barley cultivar Ganpi2 (Appendix A). According to the cloning result and further sequence analysis, it was shown that the ORF sequence of *HvLRR_8-1* is 3474 bp, which codes for a 1158 amino acid protein; the total number of atoms is 17748; the molecular formula is C_5591_H_8957_N_1483_O_1672_S_45_; the aliphatic index is 108.36; and it has a corresponding molecular weight (MW) of 125.147 kDa and a theoretical isoelectric point (pI) of 6.33. The instability index was 35.25, which was a stable protein. The percentages of basic amino acids and acidic amino acids were 7.6% and 12.7%, respectively. The predictions of transmembrane structural domains show HvLRR_8-1 has two transmembrane structural domains at 21–43 aa and 785–803 aa (Figure 4A). Secondary structure analysis of the HvLRR_8-1 protein, identified using the SOPMA program, revealed the following: 37.48% alpha helices, 11.66% extended strands, 0% beta turns, and 50.86% random coils. And the three-dimensional structure of HvLRR_8-1 was constructed using homology modeling in SWISS-MODEL with the wheat receptor-like kinase XA21 (A0A9R1M8W0) as the template. The resulting model exhibited 73.75% sequence similarity to the target HvLRR_8-1 protein, and the Global Model Quality Estimate (GMQE) value was 0.85, indicating high modeling reliability. And the LRR domain of HvLRR_8-1 adopted the characteristic curved molecular conformation typical of LRR repeat domains (Figure 4C). Sequence analysis confirmed that the HvLRR_8-1 contains seven leucine-rich repeat (LRR) domains (137–671 aa) and an S_TKc domain (836–1146 aa) (Figure 4B).

Using the ESPript 3.0 [55] software, multiple alignments of the amino acid sequences were performed with the LRR-like proteins from different plant species, revealing that the HvLRR_8-1 protein homology shared 81%, 72%, 69%, 66%, and 64% sequence identities with *Triticum aestivum* (XP044327120.1), *Aegilops tauschii* (XP020156907.3), *Lolium rigidum* (XP047074146.1), *Dichanthelium oligosanthes* (OEL19511.1), and *Phragmites australis* (XP062178854.1), respectively (Appendix A).

To elucidate the evolutionary relationships of the *HvLRR_8-1* gene in barley, a phylogenetic tree of LRR-like proteins was constructed using a total of 43 LRR amino acid sequences from 20 different plant species (Figure 4D). The results showed that the evolutionary tree is divided into five clades: Sub-clade-1 comprised 10 LRR-proteins, whereas sub-clade-2 comprised 8 LRR-proteins, Sub-clade-3 comprised 3 LRR-proteins, Sub-clade-4 comprised 6 LRR-proteins, and Sub-clade-1 comprised 16 LRR-proteins. The shortest evolutionary distance was *Triticum dicoccoides*, another member of the *Triticum aestivum*. *HvLRR_8-1* from *Hordeum vulgare* and the LRR-like protein from *Triticum aestivum* were positioned in a separate clade. The longest evolutionary distance was *Panicum hallii* in the Gramineous.

qRT-PCR analysis revealed dynamic expression changes in *HvLRR_8-1* following *Pg* infection. The exhibited progressively increasing relative expression levels of *HvLRR_8-1* in immune cultivar Ganpi2 embryos, whereas the susceptible cultivar Alexis showed a declining expression pattern of *HvLRR_8-1* over the infection time course. Compared with the CK control (non-inoculated treatment), in the immune cultivar Ganpi2, the relative expression level was significantly downregulated at 7 dpi and significantly upregulated at 14 dpi, with an extremely significant upregulation at 18 dpi. In the susceptible cultivar Alexis, the relative expression level showed no significant change at 7 dpi but was significantly downregulated at both 14 dpi and 18 dpi, with a slight increase at 18 dpi compared to 14 dpi. Overall, the *HvLRR_8-1* gene was expressed in both the immune cultivar Ganpi2 and the susceptible cultivar Alexis, but its response was more pronounced in Ganpi2 following *Pg* infection, suggesting that *HvLRR_8-1* may play a role in the defense response against *Pg* (Figure 4E).

### 2.7. Subcellular Localization of HvLRR_8-1 Fusion Protein in Tobacco

To determine the subcellular localization of HvLRR_8-1, a fusion protein of HvLRR_8-1 and EGFP was transiently localized in *N. Benthamian*. Confocal microscopy analysis revealed GFP-generated green fluorescence signals primarily localized to the cell membrane, suggesting that the *HvLRR_8-1* gene is localized predominantly in the cell membrane (Figure 5).

### 2.8. Barley HvLRR_8-1 Gene Is Crucial for Resistance to Pg

To determine whether *HvLRR_8-1* plays a role in *Pg* resistance in barley. A 497 bp fragment from the 5′ proximal coding region of *HvLRR_8-1* was cloned into BSMV: γ to silence *HvLRR_8-1* via BSMV-mediated VIGS. The successfully constructed BSMV: *HvLRR_8-1* silencing vector was transfected into barley seedling leaves through the leaf-rubbing inoculation method [56]. BSMV: γb-transfected barley plants were used as control. After *Pg* inoculation for 7 days, the results of qRT-PCR showed that, compared with the BSMV: γb control, the expression level of the *HvLRR_8-1* gene in barley leaves infected with BSMV: *HvLRR_8-1* decreased significantly by 36% at 48 h after inoculation. With the extension of the infection time to 120 h, the relative expression level of the gene decreased significantly by 52% (Figure 6B). Correspondingly, following inoculation with *Pg*, the lesion area on barley leaves treated with BSMV: *HvLRR_8-1* was significantly larger than that of the control BSMV: γb and Mock (Figure 6C). Measurement of relative chlorophyll content at *Pg* inoculation sites showed significantly lower levels in the BSMV: *HvLRR_8-1* compared to both Mock and BSMV: γb controls (Figure 6D). To further evaluate the tissue damage in *Pg*-inoculated barley leaves, Trypan Blue staining [57] was performed on the infected leaves. Following staining and decolorizing with chloral hydrate, leaves infected with BSMV: *HvLRR_8-1* exhibited more severe damage compared to the BSMV: γb and mock controls, with no staining observed in the control group leaves. The same results were observed after DAB staining. In barley leaves infected with BSMV: *HvLRR_8-1*, a significant accumulation of ROS was detected at *Pg* infection sites, whereas in the controls BSMV: γb and Mock, minimal ROS was observed, with barely detectable reactive oxygen species production (Figure 6A).

## 3. Discussion

BLS is a serious disease caused by *Pg*. Several studies on BLS have been published, covering topics such as the *Pg* genome sequence [58], resistance identification [12,13], resistant genes [14,15,16], QTL locations [13,17,18,19,20], functional verification [2], and the identification of candidate genes involved in the barley-*Pg* interaction using DDRT-PCR [21]. However, the mechanism of barley resistance to *Pg* remains poorly understood. To clarify this mechanism, DEGs at various infection stages need to be identified through transcriptome sequencing. To determine the resistance spectrum of Ganpi2 against *Pg* isolates collected from different regions in China and to analyze gene expression levels in two barley varieties—highly susceptible Alexis and immune Ganpi2, which is immune to the *Pg* isolate QWC-50 *Pg* isolates of different geographical origins were used. A total of 36 RNA-Seq libraries were constructed from infected and healthy samples at three infection stages. Ganpi2 exhibited immunity to 49 isolates, except for isolate CJZ. A substantial number of DEGs were identified in both varieties at different infection stages. Further analysis of DEG expression levels and annotations between the two varieties suggested that some DEGs may be involved in resistance to *Pg* infection.

### 3.1. Ganpi2 Has Broad-Spectrum Resistance to Pg Collected from China

The *Pg* isolate QWC is one of the most virulent isolates from Gansu Province, China [59]. While Alexis is highly susceptible to QWC, with an incidence rate of 88.24%, Ganpi2 is known to be immune to six isolates: QWC, HZZ, HYH, SD, SW, and HJX-2 [24]. In this study, Ganpi2 was artificially inoculated with 28 *Pg* isolates collected from different regions of China to determine its resistance spectrum. Ganpi2 was found to be immune to 27 isolates and showed high resistance to isolate BS-13001-2. Although previous studies have demonstrated pathogenic differentiation among *Pg* isolates [20,23], and despite the use of 28 isolates in this study, Ganpi2 remained immune to most of them. This finding aligns with a previous report that documented Ganpi2’s resistance to six *Pg* isolates from various regions of Gansu Province [24]. These results suggest that Ganpi2 may possess broad-spectrum resistance to *Pg* isolates collected from China.

### 3.2. Resistance/Susceptibility Temporal Analysis of Barley Embryo Responses to Pyrenophora graminea Infection

Based on Anita et al. [3], microscopic observation and histochemical staining techniques were employed to study barley embryo responses to *Pg* infection, using comparative analysis of near-isogenic lines resistant to NIL3876 and susceptible to Mirco. They examined 7, 11, 14, 18, 20, 24, and 27 days seven time points post-inoculation to characterize the spatiotemporal dynamics of infection. At 7 dpi of inoculation, *Pg* was detected only in the distal provascular tissue of some embryos, indicating localized early infection. By 14 dpi, *Pg* hyphae in resistant NIL3876 embryos remained confined to a small area of provascular tissue, while in susceptible Mirco embryos, hyphae had spread to the scutellum and exhibited intercellular growth without causing host cell necrosis. At 18 dpi, hyphae in resistant embryos were still restricted to the distal scutellum, whereas in susceptible embryos, *Pg* had colonized the entire scutellum and began intensive colonization of the shoot apex from 20 dpi onward. These findings established T7, T14, and T18 as critical time points for analyzing *Pg* invasion patterns in barley embryos. In our study, we similarly selected T7, T14, and T18 for transcriptomic analysis of *Pg*-resistant and susceptible barley embryos. Differential gene expression analysis revealed distinct patterns: in Alexis, comparison of *Pg*-infected versus control embryos showed 801, 1032, and 1589 upregulated DEGs and 549, 866, and 466 downregulated DEGs at 7 dpi, 14 dpi, and 18 dpi, respectively. In Ganpi2, we detected 867, 1191, and 1118 upregulated DEGs and 328, 491, and 1107 downregulated DEGs at the same time points. Notably, the substantial transcriptional changes observed in Ganpi2 at 14 dpi suggest this time point may represent a critical phase for *Pg* resistance responses, further validating the selection of T7, T14, and T18 as critical time points for investigating the *Pg*-barley interaction.

### 3.3. Plant-Pathogen Interaction Related to Defense Responses to Pg

To prevent further infection, plants activate PAMP-triggered immunity (PTI) and effector-triggered immunity (ETI) in response to pathogen invasion [60,61]. Genes related to PTI and ETI were detected during *Pg* infection, suggesting that both forms of immunity are involved in resistance to *Pg*.

Previous studies have shown that Ca^2+^ participates in plant immunity against pathogen infection by activating various downstream responses [62]. Of the 27 genes assigned to plant-pathogen interaction, two calmodulin (CaM) genes and three calmodulin-like protein genes, which serve as Ca^2+^ sensors in the pathogen response signaling cascade, were up-regulated in Ganpi2 compared to Alexis at 14 d (a critical period of Ganpi2 immunity to *Pg*), consistent with previous findings in *Brassica rapa* after infection by *Plasmodiophora brassicae* [63] and in broccoli infected by *P. brassicae* [64]. The up-regulation of CaM expression in response to *Pg* infection suggests that CaM is involved in the resistance mechanism, aligning with previous reports demonstrating CaM’s role in plant defense [65,66]. Receptor-like kinases (RLKs) present in the cytoplasmic membrane can activate PTI by recognizing pathogen-associated molecular patterns (PAMPs) [67]. Most RLK-related genes, identified among the plant-pathogen interaction-related genes, were up-regulated in both Alexis and Ganpi2; however, their expression peaked at 18 days in Alexis and at 14 days in Ganpi2, except for *HORVU2Hr1G118010*, which showed no change in Alexis but was up-regulated at 14 and 18 days in Ganpi2, with the highest expression at 18 days. These RLKs likely play a crucial role in barley’s innate immunity to *Pg*. Additionally, the salicylic acid (SA) pathway, which is induced in plants infected by hemibiotrophic pathogens [68], was implicated in resistance to *Pg*, as indicated by the up-regulation of four pathogenesis-related protein genes following *Pg* invasion.

### 3.4. Resistance Genes Participated in Response to Pg Infection

Resistance genes were identified using DIAMOND among the 457 DEGs. Although 82 DEGs were annotated as resistance genes, only 13 were derived from barley: six RLK-GNK2, three classified as “other,” two NLs, one N, and one RLP. Given that RLK-GNK2 (kinases with an additional antifungal protein ginkbilobin-2 domain) represented the majority, it appears that RLKs play a key role in the response to *Pg*, consistent with previous studies [69,70]. The *Rdg2a* gene is localized on the short arm of chromosome 7H [2]. It has been reported that Ganpi2’s resistance to *Pg* is conferred by a dominant single gene localized on the short arm of chromosome 7H, tentatively named *Rdg3* [16]. Given the small genetic distance between these loci, the possibility that *Rdg2a* and the gene conferring resistance to *Pg* are the same cannot be ruled out. Two homologs of *Rdg2a*, *HORVU7Hr1G002260* and *HORVU7Hr1G002290*, were detected among the 111 DEGs annotated as resistance genes and were expressed only in Ganpi2, not in Alexis. Because the identity and expression level of *HORVU7Hr1G002290* were higher than those of *HORVU7Hr1G002260*, we infer that *HORVU7Hr1G002290* may confer resistance to *Pg* in Ganpi2 and could be considered an allele of *Rdg2a*.

### 3.5. HvLRR_8-1 Gene Potentially Involved in Defense Response Against Pg Infection

Leucine-rich repeat receptor-like kinases (LRR-RLKs) represent a major subclass of plant receptor kinases that play pivotal roles in both plant development and immune responses. The serine/threonine kinase catalytic (S_TKc) domain serves as the core catalytic module of RLKs, mediating immune signal transduction through phosphorylation events. Notably, nearly all plant RLKs contain conserved serine/threonine phosphorylation residues essential for their signaling functions. In this study, we report the first cloning and characterization of *HvLRR_8-1*, a novel gene encoding an LRR-RLK protein kinase from the *Pg*-immune barley cultivar Ganpi2. Sequence analysis revealed that the encoded protein exhibits typical LRR-RLK structural features: an extracellular domain (137–671 aa) containing seven leucine-rich repeats (LRRs) and an intracellular domain (836–1146 aa) harboring a complete serine/threonine kinase catalytic (S_TKc) domain. Notably, although the number of LRR repeats in *HvLRR_8-1* is significantly less than that of some classical LRR-RLK proteins, such as Arabidopsis *FLS2* containing 28 LRRs [71], structural predictions suggest that these LRRs may still possess ligand-recognition ability by forming specific spatial conformations [72], which demonstrated that receptor proteins containing a small number of LRR repeats can still function by interacting with co-receptors.

Virus-induced gene silencing (VIGS) of *HvLRR_8-1* in Ganpi2 significantly compromised resistance to *Pyrenophora graminea*, as evidenced by substantially enlarged lesions, severe cellular necrosis, and a significant enrichment of ROS. These results not only confirm the critical role of *HvLRR_8-1* in barley resistance to barley leaf stripe but also provide new clues to understand the functional mechanism of type LRR-RLK containing fewer LRR repeats in plant immunity. This coincides with the finding of some research scholars that certain RLK proteins containing a small number of LRRs can be involved in immune signaling through the formation of heterodimers [73].

Of particular interest, *HvLRR_8-1* silencing led to marked ROS accumulation in barley leaves, suggesting that this LRR-RLK may additionally function in ROS homeostasis regulation. The disruption of this regulatory mechanism upon gene silencing potentially underlies the observed ROS overaccumulation. Our work represents the first functional elucidation of an LRR-RLK gene, *HvLRR_8-1*, in the barley-*Pg* interaction, providing new molecular insights into BLS resistance mechanisms.

## 4. Materials and Methods

### 4.1. Fungal Strains, Plant Material, Inoculation, and Symptom Observation

The *Pg* isolate QWC is a highly virulent strain predominantly isolated from naturally infected barley in various regions of the Hexi Corridor in Gansu, China [59]. Several *Pg* isolates collected from different regions of China, including 28 isolates used for resistance identification (Table 1) and isolate QWC, were preserved in test tubes containing hyphae covered with sterile paraffin oil.

Two barley varieties, Ganpi2 and Alexis, were used in this study. Previous research demonstrated that Ganpi2’s immunity to *Pg* isolate QWC is controlled by a single gene, temporarily designated as *Rdg3*. In contrast, Alexis was found to be highly susceptible to QWC [24].

The “sandwich” inoculation method described by Pecchioni et al. [19] was employed. Hyphae of *Pg* isolate QWC were taken from the test tube containing sterile paraffin oil and grown on potato dextrose agar (PDA, Solarbio, P8931, Beijing, China) supplemented with 13 mg/L kanamycin sulfate at 22 °C in darkness for 7 days. A 5-mm PDA plug with actively growing mycelium was then transferred to fresh PDA and incubated under the same conditions for another 7 days to prepare for inoculation. Before inoculation, the barley seeds were surface sterilized using a procedure that involved immersion in 70% ethanol for 30 s, followed by 5% sodium hypochlorite for 5 min, multiple rinses with sterile distilled water, and thorough drying.

The sterilized seeds of Ganpi2 and Alexis were placed between two layers of PDA colonized by actively growing mycelium and incubated at 6 °C for inoculation. Seeds placed between two layers of PDA without mycelium at 6 °C served as controls. Embryos were excised from the seeds after 7 (T7), 14 (T14), and 18 (T18) days of inoculation. Corresponding healthy embryos were immediately frozen in liquid nitrogen and stored at −80 °C for RNA extraction.

Conidia were scraped from infected leaves and observed by microscope (Leica, DFC450, Wetzlar, Germany). scanning electron microscope (JEOL Ltd.; JSM-5600LV, Tokyo, Japan) was used to observe the surface symptoms of infected leaves.

### 4.2. RNA Extraction, mRNA Library Construction, and Sequencing

Total RNA was extracted from both infected and healthy embryos of Alexis and Ganpi2 at T7, T14, and T18. The extraction and purification of RNA from the embryos were performed using ethanol precipitation and CTAB (BBI, CB0108-100G, Cardiff, UK) PBIOZOL (Bioer, 200 mL, Hangzhou, China) reagents following the manufacturer’s instructions. For dissociating nucleoprotein complexes, approximately 80 mg of embryos were ground in liquid nitrogen and transferred into 1.5 mL of preheated 65 °C CTAB-PBIOZOL reagent. The mixture was incubated in a Thermo mixer at 65 °C for 15 min. After incubation, the supernatant was collected by centrifugation at 12,000× *g* for 5 min at 4 °C. Then, 400 µL of chloroform was added per 1.5 mL of CTAB-PBIOZOL reagent. The supernatant was collected again by centrifuging at 12,000× *g* for 10 min at 4 °C and transferred to a new 2 mL tube. Next, 700 µL of acidic phenol and 200 µL of chloroform were added, followed by centrifugation at 12,000× *g* for 10 min at 4 °C. An equal volume of chloroform was then added, and the mixture was centrifuged under the same conditions. The supernatant was carefully aspirated and mixed with an equal volume of isopropyl alcohol. To precipitate RNA, the mixture was kept at −20 °C for 2 h. The RNA precipitate was collected by centrifugation and washed with 1 mL of 75% ethanol, then air-dried and dissolved in 50 µL of DEPC-treated water. The quality of the total RNA was assessed using a NanoDrop 8000 (Thermo Fisher Scientific, Waltham, MA, USA), and its concentration was quantified using an Agilent 2100 bioanalyzer (Agilent Technologies, Santa Clara, CA, USA).

A total of 36 libraries were constructed and sequenced, designated as A_7d_T, A_14d_T, and A_18d_T (A represents the embryos taken from Alexis, T represents the treatment group infected by *Pg*, 7d, 14d and 18d, respectively, represent the 7th day, 14th day and 18th day of *Pg* infecting the embryos of Alexis); A_7d_CK, A_14d_CK, and A_18d_CK (A represents the embryos taken from Alexis, CK represents the control group uninfected, 7d, 14d and 18d respectively represent the 7th day, 14th day and 18th day embryos of Alexis); G_7d_T, G_14d_T, and G_18d_T (G represents the embryos taken from Ganpi2, T represents the treatment group infected by *Pg*, 7d, 14d and 18d respectively represent the 7th day, 14th day and 18th day of *Pg* infecting the embryos of Ganpi2); and G_7d_CK, G_14d_CK, and G_18d_CK (G represents the embryos taken from Ganpi2, CK represents the control group uninfected, 7d, 14d and 18d respectively represent the 7th day, 14th day and 18th day embryos of Ganpi2). Three biological replicates were prepared for each treatment, named, for example, A_7d_CK-1, A_7d_CK-2, and A_7d_CK-3 for the three replicates of A_7d_CK; the other treatments were consistent with these.

To fragment the mRNA into small pieces, a fragment buffer was used on the mRNA enriched by oligo(dT)-attached magnetic beads at an optimal temperature. Based on the synthesis of the first and second cDNA strands, RNA index adapters were added to both strands by incubation with an A-tailing mix. PCR amplification was then performed on the cDNA fragments, and the amplified products were purified using Ampure XP Beads (Beckman Coulter, Brea, CA, USA) and validated using an Agilent Technologies 2100 bioanalyzer (Agilent Technologies, Santa Clara, CA, USA). The final library was obtained by heat denaturation and circularization of the double-stranded PCR products. This library was then amplified with phi29 to produce DNA nanoballs (DNBs) containing over 300 copies of each molecule. Single-ended 50-base reads were generated on a BGIseq500 platform (BGISEQ-500, Shenzhen, China) after loading the DNBs into the patterned nanoarray.

### 4.3. RNA Sequencing Data Analysis

Raw sequencing reads were filtered to obtain clean reads using SOAPnuke (v1.4.0; parameters: l, 5; q, 0.5; n, 0.1) [74] and Trimmomatic (v0.36; parameters: Illuminaclip, 2:30:10; Leading, 3; Trailing, 3; Slidingwindow, 4:15; Minlen, 50) [75]. Clean reads were then aligned to the reference genome using HISAT2 (v2.1.0; parameters: --dta--phred64 unstranded--new-summary-x index-1 read_r1-2 read_r2) [51]. Subsequently, alignment to the reference coding gene set was performed with Bowtie2 (v2.2.5; parameters: -q--phred64--sensitive--dpad 0--gbar 99999999--mp 1,1--np 1--score-min L,0, -0.1-p 16-k 200) [52]. Gene expression levels were calculated using RSEM [76]. Transcript reconstruction for each sample was carried out using StringTie (v1.0.4; parameters: -f 0.3; -j 3; -c 5; -g 100; -s 10000; -p 8) [77], and the reconstruction information was integrated using Cuffmerge [78]. The integrated transcripts were then compared with the reference genome annotation using Cuff compare [79]. Differentially expressed genes (DEGs) were identified using DEGseq, with criteria set at a fold change ≥ 4 and an adjusted *p*-value ≤ 0.001 [80]. Gene expression pattern analysis was performed using the OmicShare tools, a free online platform for data analysis (http://www.omicshare.com/tools, accessed on 15 April 2021). DEGs were functionally classified based on Gene Ontology (GO; http://www.geneontology.org/, accessed on 15 April 2021) annotation and classification, and pathway classification of DEGs was conducted according to KEGG (https://www.kegg.jp/, accessed on 15 April 2021) annotation and classification. Enrichment analysis of DEGs with GO and KEGG was performed using Phyper. Open reading frames of DEGs were identified using getorf (http://emboss.sourceforge.net/apps/cvs/emboss/apps/getorf.html, accessed on 15 April 2021) and aligned to transcription factor (TF) domains from PlantTFDB (http://plntfdb.bio.unipotsdam.de/v3.0/, current version available at: PlantRegMap http://plantregmap.cbi.pku.edu.cn/, accessed on 28 June 2025) using hmmsearch [81]. Resistance genes were predicted by aligning sequences with the PRGdb database (http://www.prgdb.org, accessed on 15 April 2021) [82] using DIAMOND (https://github.com/bbuchfink/diamond/wiki, accessed on 15 April 2021) [53].

### 4.4. Quantitative PCR (qRT-PCR) Validation of Gene Expression Data

To validate *Pg*-infected unigene expression, quantitative real-time PCR (qRT-PCR) was used. A total of twelve key genes in the plant-pathogen interaction pathway were selected. Primers for qRT-PCR were designed using Primer Express 3.0, with the amplification product size set between 70 and 150 bp. Barley *HvUBIQUITIN* was used as the internal control [83]. All primers for qRT-PCR are listed in Additional File: Appendix A. Reverse transcription was performed using the PrimeScript™ RT reagent Kit with gDNA Eraser, following the manufacturer’s instructions, with 2 µg of total RNA. qRT-PCR was conducted on an ABI ViiA 7 Real-Time PCR System (Applied Biosystems, Carlsbad, CA, USA) using SYBR Green Mastermix (Qiagen, Germantown, MD, USA). Relative expression levels of the selected unigenes, normalized to the expression level of the internal gene, were calculated using the 2^−ΔΔCt^ method.

### 4.5. Cloning and Analysis of HvLRR_8-1

The fresh leaves of Ganpi2 were collected to extract RNA and synthesize cDNA. Using Ganpi2 cDNA as a template, the *HvLRR_8-1* CDS was PCR amplified using the primer set *HvLRR_8-1-*F and *HvLRR_8-1-*R (Table 2). The PCR reaction mix was as follows: 50 μL 2 × Phanta Max Buffer, 1 μL dNTP, 1 μL cDNA, 2 μL primers (10 μM), and 1 μL Phanta Max Super-Fidelity DNA Polymease (1 U/μL). The PCR conditions were as follows: 30 s at 95 °C for denaturation, followed by 39 cycles of 15 s at 95 °C, 15 s at 49 °C, 1 min at 72 °C, and 72 °C for 5 min for amplification. The PCR amplification products were gel purified, cloned, and transformed into competent *E. coli* DH5α cells. The positive clones were selected through blue-white spot screening and PCR verification, followed by sequencing at Sangon Biotech (Shanghai, China).

The online BLAST software of the NCBI (National Center for Biotechnology Information) (https://blast.ncbi.nlm.nih.gov/, accessed on 28 June 2025) database was used to analyze DNA and protein sequences. The physicochemical properties of the protein were analyzed by Expasy Protparam (http://web.expasy.org/protparam/, accessed on 18 June 2025). Transmembrane structural domain prediction for this protein used online software TMHMM 2.0 (https://services.healthtech.dtu.dk/services/TMHMM-2.0/, online software accessed on 24 May 2025). The secondary structure and tertiary structure of the HvLRR_8-1 protein were predicted using the SOPMA program (https://npsa.lyon.inserm.fr/cgi-bin/secpred_sopma.pl, accessed on 18 June 2025) and the SWISS-MODEL (https://swissmodel.expasy.org/, accessed on 16 July 2025). Annotation of HvLRR_8-1 protein structural domains using the SMART (http://smart.embl-heidelberg.de/, accessed on 18 June 2025) protein structural domain database platform. To reveal the evolutionary relationship of HvLRR_8-1 between barley and other species, multiple alignments of Leucine-rich-repeat receptor-like kinases (LRR-RLKs) proteins were carried out using the ESPript 3.0 software. We used the cloned HvLRR_8-1 sequence to search for key LRR-RLK proteins in NCBI blastp. The Neighbor-Joining (NJ) phylogenetic tree was constructed using MEGA 11 with 1000 bootstrap repeats to represent evolutionary relationships [84]. The resulting “Newick” tree file was then exported and displayed on the EvolView website (https://www.evolgenius.info/evolview-v2, accessed on 26 May 2025) (Appendix A).

The “sandwich” inoculation method was similarly used to inoculate barley with a *Pg* isolate highly virulent strain QWC, the sterilized seeds of Ganpi2 and Alexis were placed between two layers of PDA colonized by actively growing mycelium and incubated at 6 °C for inoculation. Seeds placed between two layers of PDA without mycelium served as uninfected controls CK at 6 °C. Embryos were excised from the seeds at 7 dpi, 14 dpi, and 18 dpi (7 dpi, 14 dpi, and 18 dpi represent the 7th, 14th, and 18th days post-infection). Embryos were immediately frozen in liquid nitrogen and stored at −80 °C for RNA extraction. For each treatment at each time point, three sampling replicates were set up. The qRT-PCR reaction system and procedure followed the manufacturer’s instructions of SYBR Green Mastermix (Qiagen, Germantown, MD, USA). The barley *HvUBIQUITIN* gene (NCBI accession no. AK361071, HvUBC) was used as the internal reference gene (Appendix A). Calculated HvLRR_8-1 relative expression levels across time points in different treatments by the 2^−ΔΔCt^ method. Data analysis and graphical presentation of *HvLRR_8-1* relative expression levels were performed using Origin 2022 software (OriginLab, Northampton, MA, USA).

### 4.6. Subcellular Localization

The *N. Benthamian* transient expression system was used to ascertain the subcellular localization of HvLRR_8-1. The *HvLRR_8-1* CDS was cloned into the pRI101-EGFP vector to generate a recombinant vector (pRI101-EGFP: HvLRR_8-1). The pRI101-EGFP: HvLRR_8-1 construct and pRI101-EGFP empty vector were transformed into *Agrobacterium tumefaciens* strain GV3101 for infecting the *Nicotiana benthamiana* leaves [85]. The leaves of 4-week-old tobacco plants were infiltrated and subsequently incubated in a controlled chamber at 22 °C with a 12 h light/dark cycle for 2–3 days. The distribution of the EGFP fluorescence signal in the transformed tobacco leaves was observed and photographed using a laser confocal scanning microscope with an excitation wavelength of 488 nm (Olympus FV1200, Tokyo, Japan). The primer sequences constructed for subcellular localization analysis are listed in Table 2.

### 4.7. Barley Stripe Mosaic Virus-Induced HvLRR_8-1 Gene Silencing and Plant Inoculation

Using the BSMV-VIGS method to define the *HvLRR_8-1* role in disease-related response to *Pg* infection. A 497 bp specific sequence was selected from the cDNA of the *HvLRR_8-1* gene as the target sequence of VIGS, and specific primers were designed in Table 2. Using the existing *HvLRR_8-1* recombinant plasmid as the template, the fragments were amplified by vazyme phanta Max super fidelity DNA polymerase PCR. Total PCR reaction system 50 μL: 25 μL 2× Phanta Max Buffer, 1 μL dNTP Mix (10 mM), template DNA 2 μL, a couple of primers 2 μL, 1 μL Phanta Max Super-Fidelity DNA Polymease (1U/μL). The PCR conditions were as follows: 30 s at 95 °C for denaturation, followed by 39 cycles of 15 s at 95 °C, 15 s at 48 °C, 1 min at 72 °C, Finally, it was extended at 72 °C for 5 min for amplification. After the PCR amplification products were gel purified. The T4 DNA ligase was used to connect the target fragment with the linearized γ vector (treated with restriction enzyme *APA* I), and the linked product was transformed into *E. coli*. The primer γ-*HvLRR*-F/R (Table 2) was used to screen the positive clone and sequencing. The recombinant plasmid was named γ-*HvLRR_8-1*.

The two plasmids of γ (Empty vector), and γ-*HvLRR_8-1* were respectively mixed with BSMV: αRNA and BSMV: βRNA in a 1:1:1 ratio to assemble three VIGS viruses, namely BSMV: γb, BSMV: *PDS*, and BSMV: *HvLRR_8-1*. VIGS of phytoene desaturase (PDS) induces photobleaching phenotypes. Barley inoculated with BSMV: *PDS* was used to verify whether the VIGS silencing system used was effective. Ganpi2 barley inoculated with BSMV: *HvLRR_8-1* was used as the experimental group to study the resistance of barley to *Pg* after the *HvLRR_8-1* gene was silenced. No genes were silenced in the barley inoculated with BSMV: γb as a control, and untreated wild-type barley was included as an additional mock control. Two-week-old Ganpi2 seedlings were separately inoculated with BSMV: γb and BSMV: *HvLRR_8-1*. The rub inoculation was conducted by pipetting the mixture on the downmost second leaves and by squeezing it gently with a gloved hand through the entire leaf surface from its base to the tip two to three times. Fourteen days after inoculation, photobleaching was observed in the barley leaves inoculated with BSMV: *PDS*. Then, the laboratory-preserved *Pg* isolate QWC, using a hole puncher with a diameter of 5 mm, small pathogenic bacteria cakes were punched along the edge of the QWC colonies and inoculated upside down onto the second leaf of BSMV: γb and BSMV: *HvLRR_8-1*, the barley leaves. Then covered with plastic film to maintain high relative humidity (85%) for 120 h in an illumination incubator. On the 7th day after inoculation with *Pg*, the disease phenotype of barley leaves was observed. The *Pg*-infected lesion areas barely treated with Mock, BSMV: γb, and BSMV: *HvLRR_8-1* were analyzed with ImageJ-win64 software. Simultaneously, the relative chlorophyll content at *Pg* infection sites of the corresponding treatments was measured with an SPAD chlorophyll meter (TYS-4N; Zhongke Weihe Technology Development Co., Ltd., Beijing, China). Meanwhile, to investigate the effects of silencing the *HvLRR_8-1* gene on ROS accumulation and cell death during *Pg* infection in barley leaves, both Trypan Blue staining and 3,3′-Diaminobenzidine (DAB) staining methods were applied to *Pg*-infected barley leaves, and the effect of *HvLRR_8-1* gene silencing on the disease resistance of barley was analyzed.

### 4.8. qRT-PCR

To analyze the effect of *Pg* infection on the expression level of the silenced disease-related gene *HvLRR_8-1*. On the 48 h and 120 h after inoculation with the *Pg*, the leaves of VIGS barley seedlings were sampled. Total RNA was extracted from the *HvLRR_8-1* silenced treatment and the BSMV: γb control, respectively. Using the RNA simple Total RNA Kit (TIANGEN, DP419), extract RNA, and then reverse-transcribe it into cDNA using FastKing gDNA Dispelling RT Super Mix (TIANGEN, KR118-02, Beijing, China). The transcription expression levels of *HvLRR_8-1* were detected by qRT-PCR using SYBR Premix Ex Taq (Takara Bio, Dalian, China). The specific primers used for *HvLRR_8-1* (Table 2: BSMV: *HvLRR_8-1*-F and BSMV: *HvLRR_8-1*-R) expression analysis are shown in Table 2. *Actin* was used as the internal reference gene. The *HvLRR_8-1* expression levels were quantified using the 2^−∆∆Ct^ method. Origin 2022 was used for drawing.

### 4.9. Statistical Analysis

In this study, all experiments were repeated in triplicate, and each sampling was analyzed separately. Statistical analysis was performed with SPSS 20.0. The data from the quantitative *HvLRR_8-1* assay were subjected to analysis of variance, and the mean values were compared using the LSD test (*p* = 0.05). Data were compared using two-way ANOVA, * (*p* < 0.05), ** (*p* < 0.01) and *** (*p* < 0.001) designate that the differences are statistically significant, highly significant, and extremely statistically significant, respectively.

## 5. Conclusions

*Pg* poses a significant threat to barley yield production, yet the underlying defense mechanisms of barley remain unclear. This study employed RNA-Seq to investigate the transcriptional responses in embryos of highly susceptible and resistant barley varieties at three stages of *Pg* infection. The transcript level changes in Ganpi2 occurred more rapidly compared to Alexis during the infection process. A total of 457 DEGs associated with Ganpi2’s resistance to *Pg* were identified, predominantly related to plant-pathogen interactions and kinase activation. Among these DEGs, 82 were classified as resistance genes, and 24 were transcription factors (TFs). Of the 27 genes involved in plant-pathogen interaction, eight encoding receptor-like kinases (RLKs) were identified. Of the 27 genes involved in plant-pathogen interaction, one encodes the receptor LRR S_TKc kinase protein gene. Based on integrated whole-transcriptome results, we further identified a key gene, *HvLRR_8-1*. After *HvLRR_8-1* gene was further silenced using the BSMV-VIGS method, BSMV: *HvLRR_8-1*-infected barley seedlings showed diminished disease resistance to *Pg*, indicating that the barley *HvLRR_8-1* gene was a key regulator of resistance to *Pg*. These findings offer new insights into the mechanisms of barley resistance to *Pg* and elucidate the role of Leucine-rich-repeat receptor-like kinases (LRR-RLKs) protein in barley’s response to *Pg*.

## Figures and Tables

**Figure 1 plants-14-02350-f001:**
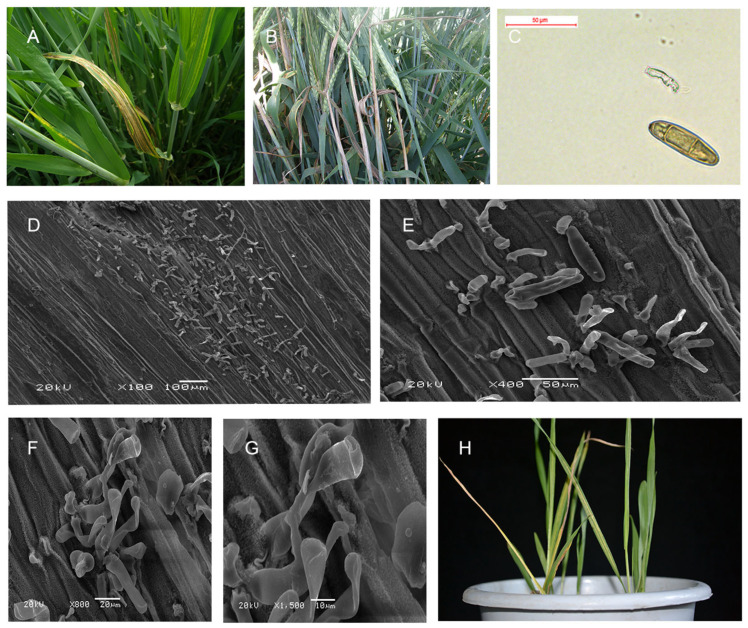
The symptom observation of barley leaf stripe. (**A**): Barley leaf stripe symptoms of a natural infection at boot stage in the field. (**B**): Barley leaf stripe symptoms of a natural infection at filling stage in the field. (**C**): Morphological observation of conidia from leaf infected with barley leaf stripe. (**D**): The symptoms observed in a scanning microscope for magnification at ×100. (**E**): The symptoms observed in a scanning microscope for magnification at ×400. (**F**): The symptoms were observed in a scanning microscope for magnification at ×800. (**G**): The symptoms observed in a scanning microscope for magnification at ×1500. (**H**): Barley leaf stripe symptoms of Alexis after an artificial inoculation at seedling stage in the phytotron.

**Figure 2 plants-14-02350-f002:**
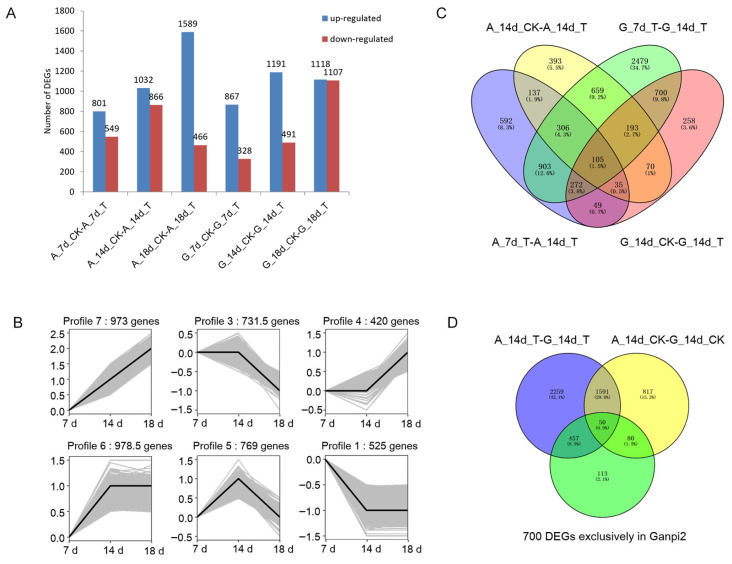
Differentially expressed gene selection related to immunity of Ganpi2 to *Pyrenophora graminea*. (**A**): Differentially expressed genes between inoculated and uninoculated barley at the same point in time. (**B**): Expression pattern differences between Alexis and Ganpi2 after *Pyrenophora graminea* infection. (**C**): Differentially expressed genes have special existence in G_7d_T-G_14d_T and G_14d_CK-G_14d_T, but not in A_7d_T-A_14d_T and A_14d_CK-A_14d_T. (**D**): Differentially expressed genes have a special existence in 700 DEGs exclusively in Ganpi2 and A_14d_T-G_14d_T, but not in A_14d_CK-G_14d_CK. A represents the embryos taken from Alexis, G represents the embryos taken from Ganpi2, T represents the treatment group infected by *Pg*, and CK represents the control group uninfected; 7d, 14d, and 18d respectively represent the 7th day, 14th day, and 18th day of *Pg* infecting the embryos.

**Figure 3 plants-14-02350-f003:**
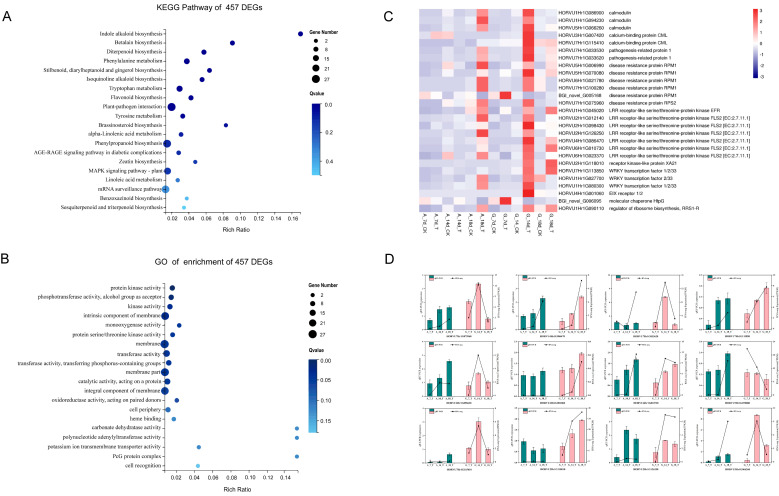
Annotation and qRT-PCR validation of differentially expressed genes related to barley immunity to *Pyrenophora graminea*. (**A**): Bubble chart of KEGG enrichment of 457 DEGs related to immunity of Ganpi2 to *Pyrenophora graminea*; (**B**): Bubble chart of GO enrichment of 457 DEGs related to immunity of Ganpi2 to *Pyrenophora graminea.* (**C**): Heatmap of 27 DEGs belonging to Plant-Pathogen interaction. Red represents up-regulation, and blue represents down-regulation of gene expression. Expression values were FPKM. (**D**): Quantitative real-time PCR validation of 12 differentially expressed genes and the corresponding expression data of RNA-Seq during various stages of barley infected by *Pyrenophora graminea*. The columns represent relative expression levels obtained by qRT-PCR, and the broken line represents relative expression obtained by RNA-Seq. Error bars represent standard errors.

**Figure 4 plants-14-02350-f004:**
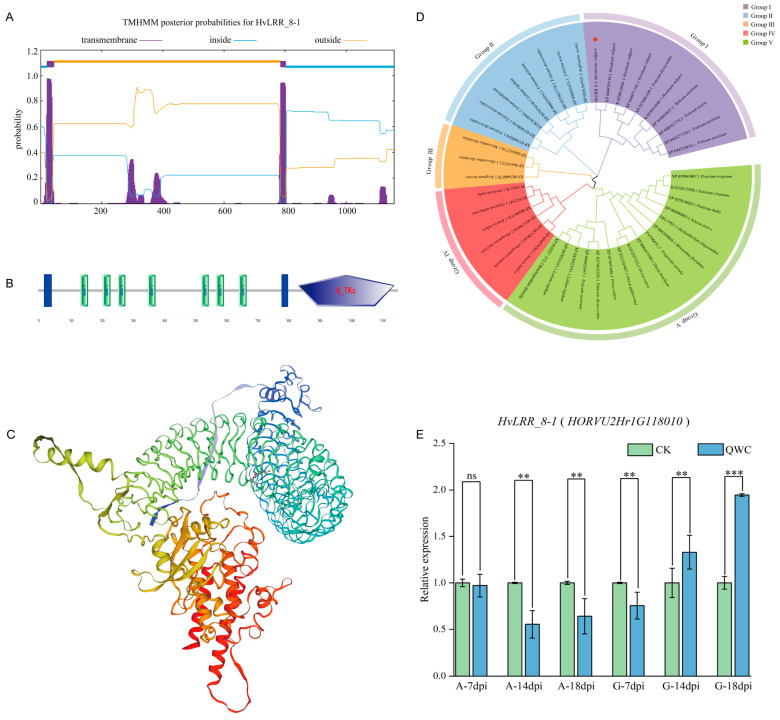
Expression analysis and bioinformatics analysis of the barley receptor protein kinase gene *HvLRR_8-1*. (**A**): Prediction of the transmembrane structure of the HvLRR_8-1 protein. (**B**): Schematic representation of the functional structural domains of the protein encoded by the *HvLRR_8-1* gene. Leucine-rich repeat (LRR), transmembrane region, and S_TKc kinase structural domains are represented by green squares, blue squares, and pentagons, respectively. (**C**): Prediction of HvLRR_8-1 protein tertiary structure. (**D**): Phylogenetic tree of barley HvLRR_8-1 and other related LRR proteins by adopting maximum livelihood model with bootstrap value *n* = 1000 using MEGA 11.0. The red dot shows the position of HvLRR_8-1. (**E**): The expression pattern of *HvLRR_8-1* gene response to *Pyrenophora graminea*. 7 dpi, 14 dpi, and 18 dpi represent 7th, 14th, and 18th days post-infection, A denotes the *Pg*-susceptible barley cultivar Alexis, and G denotes the *Pg*-immune barley cultivar Ganpi2. CK represents the uninoculated control group, and QWC represents the inoculated with the highly pathogenic *Pyrenophora graminea* strain QWC. ns: No significant difference, **: *p* < 0.05, ***: *p* < 0.001.

**Figure 5 plants-14-02350-f005:**
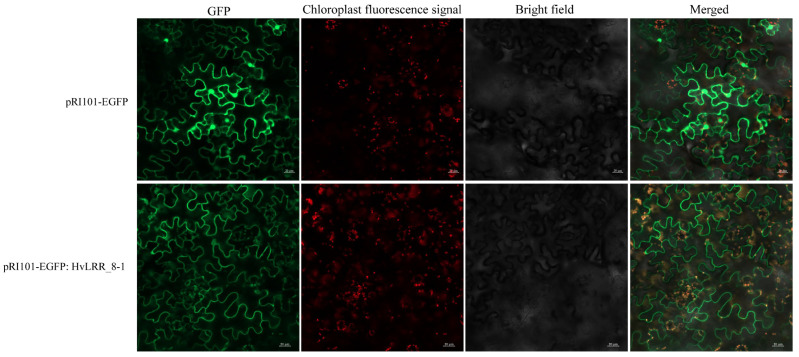
Subcellular localization of the barley HvLRR_8-1 protein in *N. benthamiana*. Green fluorescence protein (GFP), chlorophyll autofluorescence (Auto), bright-field, and merged images are shown. Scale bar represents 20 μm. pRI101-EGFP: HvLRR_8-1 fusion protein was detected in the cell membrane, and pRI101-EGFP fusion protein was used as a positive control.

**Figure 6 plants-14-02350-f006:**
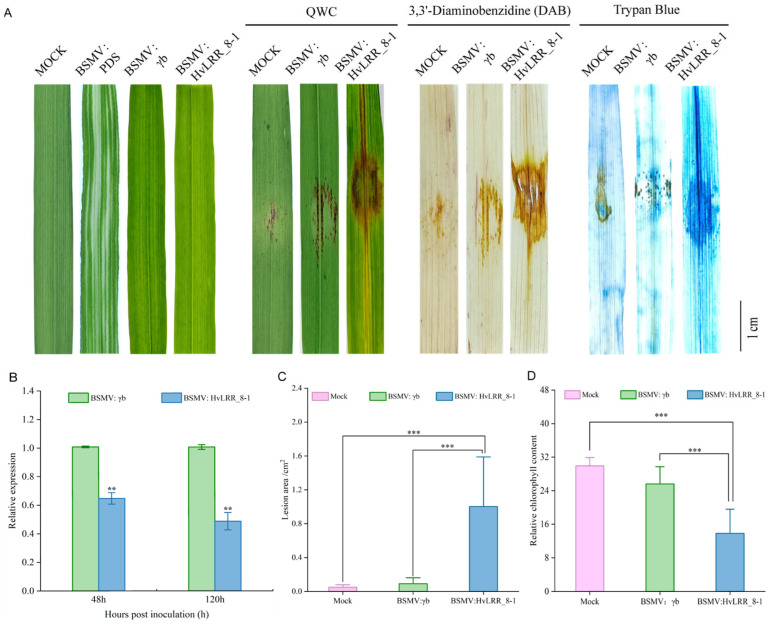
Phenotype of the *HvLRR_8-1*-silenced barley leaves and the identification of *Pyrenophora graminea* resistance. (**A**): Comparison leaves phenotype and staining symptoms of the wild-type Ganpi2 barley leaves (Mock), the vector control barley leaves (BSMV: γb), and the *HvLRR_8-1*-silenced barley leaves (BSMV: *HvLRR_8-1*) inoculated with *Pyrenophora graminea*. Scale bar represents 1 cm. (**B**): Relative expression level of *HvLRR_8-1* in BSMV: γb, and BSMV: *HvLRR_8-1* after inoculating with *Pyrenophora graminea*. (**C**): Lesion areas of *HvLRR_8-1* in Mock, BSMV: γb, and BSMV: *HvLRR_8-1* after inoculating with *Pyrenophora graminea*. (**D**): Relative chlorophyll content (SPAD) of *HvLRR_8-1* in Mock, BSMV: γb, and BSMV: *HvLRR_8-1* after inoculating with *Pyrenophora graminea*. **: *p* < 0.01, ***: *p* < 0.001.

**Table 1 plants-14-02350-t001:** Resistance identification of Ganpi2 to isolates collected from different regions in China.

Isolate Name	Geographic Origin	Disease Incidence (%)	Reaction Types	Isolate Name	Geographic Origin	Disease Incidence (%)	Reaction Types
BS-14095-2-1	Hefei, Anhui	0.00	I	BS-13007-7-1	Eryuan, Dali, Yunnan	0.00	I
BS-13032-2	Hohhot, Inner Mongolia	0.00	I	BS-13009-8-1	Eryuan, Dali, Yunnan	0.00	I
BS-13032-3	Hohhot, Inner Mongolia	0.00	I	BS-13003-1	Jianchuan, Dali, Yunnan	0.00	I
BS-13020-1	Huzhu, Haidong, Qinghai	0.00	I	BS-13003-2	Jianchuan, Dali, Yunnan	0.00	I
BS-13019-4	Huangzhogn, Xining, Qinghai	0.00	I	BS-13002-3	Jianchuan, Dali, Yunnan	0.00	I
BS-13014-5	Menyuan, Haibei, Qinghai	0.00	I	BS-13002-2	Jianchuan, Dali, Yunnan	0.00	I
BS-13014-2	Menyuan, Haibei, Qinghai	0.00	I	BS-13004-2-1	Jianchuan, Dali, Yunnan	0.00	I
BS-13015-1-2	Menyuan, Haibei, Qinghai	0.00	I	BS-13004-2-3	Jianchuan, Dali, Yunnan	0.00	I
BS-13016-3	Menyuan, Haibei, Qinghai	0.00	I	BS-13004-4-2	Jianchuan, Dali, Yunnan	0.00	I
BS-13018-3-4	Haiyan, Haibei, Qinghai	0.00	I	BS-13004-3-1	Jianchuan, Dali, Yunnan	0.00	I
BS-13018-1-5	Haiyan, Haibei, Qinghai	0.00	I	BS-13004-1-1	Jianchuan, Dali, Yunnan	0.00	I
BS-13001-2	Haiyan, Haibei, Qinghai	2.56	HR	BS-13001-2	Jianchuan, Dali, Yunnan	0.00	I
BS-13055-2-3	Dechang, Liangshan, Sichuan	0.00	I	BS-13012-6-5	Panlong, Kunming, Yunnan	0.00	I
BS-13055-2-1	Dechang, Liangshan, Sichuan	0.00	I	BS-13012-6-1	Panlong, Kunming, Yunnan	0.00	I

Note: I: Immune; HR: highly resistant.

**Table 2 plants-14-02350-t002:** Primers information used in the study.

Code	Primer	Sequence (5′–3′)	Purpose
1	*HvLRR_8-1-*F	ATGTGTGATAGAAAGCACATG	Cloning *HvLRR_8-1* gene
2	*HvLRR_8-1-R*	GCTGTGCAACGCTGCAAATG	
3	35S-F	GACGCACAATCCCACTATCC	Clone *HvLRR_8-1* to EGFP for subcellular localization assay in *N. benthamiana*
4	GFP-JR	GGGTCAGCTTGCCGTAGGTG	
5	γ-*HvLRR*-F	TTGCTCCTAACATCAATACC	Clone HvLRR_8-1 gene silencing fragment was used for the construction of the BSMV-VIGS clone. (The red font represents the homologous recombination arm of the connection carrier)
6	γ-*HvLRR*-R	CCCAGCAAGTATTGGTAAGC	
7	BSMV: *HvLRR*-F	CGCTTCCAAGCATCCAAACTT	The primers for the *HvLRR_8-1* gene were used for qRT-PCR detection.
8	BSMV: *HvLRR*-R	GGCAAGCGAAGTGGGAATTT	
9	Actin-F	GCTGACCGTATGAGCAAGGA	Expression level analysis of internal control gene *Actin* in barley
10	Actin-R	GGAAAGTGCTGAGTGAGGCT	

## Data Availability

Data are contained within the article and Appendix A.

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
