# Peer review of "Transcriptome Analysis and Functional Characterization of the HvLRR_8-1 Gene Involved in Barley Resistance to Pyrenophora graminea"

_plants, 2025, doi:10.3390/plants14152350_

Round 1
Reviewer 1 Report
Comments and Suggestions for Authors
In this study, authors employed transcriptome analysis and experimental evidence to uncover the gene involved in barley resistance to Pyrenophora graminea and suggested HvLRR_8-1 played a critical role in regulating barley resistance. The design and results were acceptable. Whereas I wonder that does HvLRR_8-1 uniquely existence in immune Ganpi2 cultivar? And does this gene existence in highly susceptible Alexis and other HR, MR, and MS cultivars? How about its expression patterns in those cultivars. Besides, authors used 7, 14, and 18 days’ samples to perform the transcriptome analysis. The reason for choosing this time points should be discussed in Discussion section to self-prove this time points are suitable for the study purpose. Other points are listed following.
L32. In “expressed on the cell membrane”, expressed is not proper. As usually “expressed” used for description gene, and protein was usually description as localization. Also, in L338.
The “Introduction” section seems incomplete and miss the introduction to the usage of transcriptome analysis in revealing the resistance mechanisms and mining the resistance genes, which is the core work of this study.
L193. It is suggested give the definition of each abbreviation. take G_7d_T-G_14d_T as an example, readers may cannot follow G and T represent for when just from the figure itself.
L215. Pg need italic or not?
L299. 2.6. Cloning and Characterization of HvLRR_8-1 Gene. It is suggested authors add characterization analysis of HvLRR_8-1 including 3D structure, transmembrane status, the induced expression patterns to merge with Figure 4. Only a phylogenetic tree is too simple.
L321-330. Several Latin name should be italic. Double check throughout the manuscript.
L331. In Figure 4, the Latin name should be italic.
L346. The “we silenced 497 bp fragment” is strange description.
L348. It seems the sentience “Constructed a BSMV: HvLRR_8-1 silencing vector, which was inoculated into the leaves of barley seedlings.” is not a complete sentence.
L367. The result of Figure 6 D is not mentioned in Result section. Meanwhile, SPAD is not defined.
Comments on the Quality of English LanguageSuch as the sentience “Constructed a BSMV: HvLRR_8-1 silencing vector, which was inoculated into the leaves of barley seedlings.” is not a complete sentence.
Author Response
Dear Editors and Reviewers:
Thank you for your letter and for the reviewers' comments concerning our manuscript entitled “Transcriptome analysis and functional characterization of the HvLRR_8-1 gene involved in barley resistance to Pyrenophora graminea” (lD: plants-3756417). Those comments are all valuable and very helpful for revising and improving our paper, as well as the important guiding significance to our research. We have studied comments carefully and have made corrections which we hope to meet with approval. Revised portions are marked in red in the manuscript. The main corrections in the manuscript and the responds to the reviewer's comments are as flowing:
Responds to the reviewer's comments:
Reviewer #1
In this study, authors employed transcriptome analysis and experimental evidence to uncover the gene involved in barley resistance to Pyrenophora graminea and suggested HvLRR_8-1 played a critical role in regulating barley resistance. The design and results were acceptable. Whereas I wonder that does HvLRR_8-1 uniquely existence in immune Ganpi2 cultivar? And does this gene existence in highly susceptible Alexis and other HR, MR, and MS cultivars? How about its expression patterns in those cultivars. Besides, authors used 7, 14, and 18 days’ samples to perform the transcriptome analysis. The reason for choosing this time points should be discussed in Discussion section to self-prove this time points are suitable for the study purpose. Other points are listed following.
[Response] We thank you for your encouraging comments on our manuscript. After carefully considering all the comments, we have performed the point-by-point revisions. Based on the previous phenotypic research results, this study focused on two extreme barley disease-resistant varieties as the main materials for research. We only conducted expression pattern analysis in the immune variety Ganpi2 and the highly susceptible variety Alexis and found that there was indeed a significant difference in the expression trend of HvLRR_8-1 gene between the two extreme varieties. Therefore, it was determined that this gene would be used for functional research. 3.2. Resistance/Susceptibility Temporal Analysisof Barley Embryo Responses to Pyrenophora graminea Infection. [line , Figure 4 E]. Additionally, the selection of T7, T14, and T18 samples for transcriptome analysis was primarily based on the research findings of Anita et al. (citation [3] in References). And we have included relevant discussion in the Discussion section regarding both the reasons for selecting these time points and the corresponding results. The revised text reads as follows: Based on Anita et al. [3] employed microscopic observation and histochemical staining techniques to study barley embryo responses to Pg infection, using comparative analysis of near-isogenic lines resistant NIL3876 and susceptible Mirco. They examined 7, 11, 14, 18, 20, 24, and 27 days seven time points post-inoculation to characterize the spatiotemporal dynamics of infection. At 7dpi of inoculation, Pg was detected only in the distal provascular tissue of some embryos, indicating localized early infection. By 14 dpi, Pg hyphae in resistant NIL3876 embryos remained confined to a small area of provascular tissue, while in susceptible Mirco embryos, hyphae had spread to the scutellum and exhibited intercellular growth without causing host cell necrosis. At 18dpi, hyphae in resistant embryos were still restricted to the distal scutellum, whereas in susceptible embryos, Pg had colonized the entire scutellum and began intensive colonization of the shoot apex from 20dpi onward. These findings established T7, T14, and T18 as critical time points for analyzing Pg invasion patterns in barley embryos. In our study, we similarly selected T7, T14, and T18 for transcriptomic analysis of Pg-resistant and susceptible barley embryos. Differential gene expression analysis revealed distinct patterns: in Alexis, comparison of Pg-infected versus control embryos showed 801, 1,032, and 1,589 upregulated DEGs and 549, 866, and 466 downregulated DEGs at 7dpi, 14dpi, and 18dpi, respectively. In Ganpi2, we detected 867, 1,191, and 1,118 upregulated DEGs and 328, 491, and 1,107 downregulated DEGs at the same time points. Notably, the substantial transcriptional changes observed in Ganpi2 at 14dpi suggest this time point may represent a critical phase for Pg resistance responses, further validating the selection of T7, T14, and T18 as critical time points for investigating the Pg-barley interaction. (line 500-523, red words) Please find our responses to other comments/suggestions listed below.
Major points:
- L32. In “expressed on the cell membrane”, expressed is not proper. As usually “expressed” used for description gene, and protein was usually description as localization. Also, in L338.
[Response 1] We thank the reviewer for this suggestion. We re-written this sentence according to the Reviewer’s suggestion. The revised text reads as follows: Subcellular localization in N. Benthamian indicates that the HvLRR_8-1 is localized on the cell membrane. (line 32, red words). To determine the subcellular localization of HvLRR_8-1, a fusion protein of HvLRR_8-1 and EGFP was transiently localized in N. Benthamian. (line 414, red words).
- The “Introduction” section seems incomplete and miss the introduction to the usage of transcriptome analysis in revealing the resistance mechanisms and mining the resistance genes, which is the core work of this study.
[Response 2] We thank the reviewer for this suggestion. We re-written this sentence according to the Reviewer’s suggestion. With the advancement of RNA-seq, it has been widely used to identify resistance factors in various plants infected by pathogens. and has become an essential tool for deciphering plant-pathogen interactions [26]. Similarly, a substantial repertoire of genes controlling critical disease-resistance traits in barley has been characterized through RNA-seq analyses. Zeng et al. [27] identified 831 powdery mildew-responsive differentially expressed genes in Tibetan barley using RNA-seq. Wang et al. [28] iden-tified four pairs associated with barley leaf stripe miRNAs and target genes through RNA-seq, combined with miRNA-seq and degradation-seq: The differential expression of Hvu-miR168-5p and HvAGO1, Hvu-novel-52 and HvGRF6, Hvu-miR6195 and CLP, Hvu-miR159b and GAMYB, triggered the expression of key target genes. Similarly, transcriptome analysis of barley leaf spot, caused by Ramularia collocygni, demonstrat-ed the critical role of xylanase inhibitors during early pathogen recognition [29]. In re-cent years, transcriptomic sequencing has also been widely used to analyze changes in genes expression related to the barley-pathogen interaction and has also proven to be pivotal in identifying core pathways associated with disease resistance in barley. Mehtap et al [30]. Integrated transcriptomic profiling in the Bipolaris sorokiniana resistant barley cultivar Dara revealed significant upregulation of the PR3 gene, highlighting the role of chitinase-mediated cell wall reinforcement in defense. This resistance mechanism functions through degradation of the invading pathogen's cell wall com-ponents. Additionally, Ghannam et al. [31] employed RNA-seq to identify 15 barley key disease-resistant genes potentially involved in Pg-immune responses, while also revealing that oxidative metabolic processes such as reactive oxygen species (ROS)-related pathways may contribute to disease resistance mechanisms. In barley resistant to powdery mildew Blumeria graminis, PR1 and PR5 in the salicylic acid (SA) pathway, were significantly upregulated, while the jasmonic acid (JA) pathway appeared to be suppressed by pathogen effector proteins [32]. Another study demon-strated that reactive oxygen species (ROS) metabolism,spot blotch-resistant cultivars rapidly activated the ROS scavenging system to restrict pathogen proliferation [33]. (line 75-101, red words).
- L193. It is suggested give the definition of each abbreviation. take G_7d_T-G_14d_T as an example, readers may cannot follow G and T represent for when just from the figure itself.
[Response 3] We thank the reviewer for this suggestion. The main areas where we made modifications are in legend of Figure 2: A represents the embryos taken from Alexis, G represents the embryos taken from Ganpi2, T represents the treatment group infected by Pg, CK represents the control group uninfected; 7d, 14d and 18d respectively represent the 7th day, 14th day and 18th day of Pg infecting the embryos. (line226-229, red words). And the revised text reads as follows: A_7d_T, A_14d_T, and A_18d_T (A represents the embryos taken from Alexis, T represents the treatment group infected by Pg, 7d, 14d and 18d respectively represent the 7th day, 14th day and 18th day of Pg infecting the embryos of Alexis); A_7d_CK, A_14d_CK, and A_18d_CK (A represents the embryos taken from Alexis, CK represents the control group uninfected, 7d, 14d and 18d respectively represent the 7th day, 14th day and 18th day embryos of Alexis); G_7d_T, G_14d_T, and G_18d_T (G represents the embryos taken from Ganpi2, T represents the treatment group infected by Pg, 7d, 14d and 18d respectively represent the 7th day, 14th day and 18th day of Pg infecting the embryos of Ganpi2); and G_7d_CK, G_14d_CK, and G_18d_CK (G represents the embryos taken from Ganpi2, CK represents the control group uninfected, 7d, 14d and 18d respectively represent the 7th day, 14th day and 18th day embryos of Ganpi2). (line 647-658, red words).
- L215. Pg need italic or not?
[Response 4] we sincerely thank the reviewer for pointing out the problem. As suggested by the reviewer, we have corrected line 545, line 448, line 426, line 574, line 759, line 762.
- L299. 2.6. Cloning and Characterization of HvLRR_8-1 Gene. It is suggested authors add characterization analysis of HvLRR_8-1 including 3D structure, transmembrane status, the induced expression patterns to merge with Figure 4. Only a phylogenetic tree is too simple.
[Response 5] We thank the reviewer for this suggestion. According to the reviewer’s advice, we have added Figures 4A: Prediction of the transmembrane structure of the HvLRR_8-1 protein, 4B: Schematic representation of the functional structural domains of the protein encoded by the HvLRR_8-1 gene. Leucine-rich repeat (LRR), transmembrane region and S_TKc kinase structural domains are represented by green squares, blue squares and pentagons, respectively., 4C: Prediction of HvLRR_8-1 protein tertiary structure. And 4E: The expression pattern of HvLRR_8-1 gene response to Pyrenophora graminea to the relevant part of Figure 4. (line 388-416, red word).
- L321-330. Several Latin name should be italic. Double check throughout the manuscript.
[Response 6] We sincerely thank the reviewer for careful reading. As suggested by the reviewer, we have carefully reviewed the entire text and have made corrections to all the parts that did not use italic font. These changes in font position are respectively located at line 242, line 381, line 448, line 717 (E. coli), line 721 (HvLRR_8-1).
- L331. In Figure 4, the Latin name should be italic.
[Response 7] Thanks for your careful checks. We feel sorry for our carelessness. The Latin names in Figure 4 E have been uniformly indicated in italic. (line 402, Figure 4)
- L346. The “we silenced 497 bp fragment” is strange description.
[Response 8] We thank the reviewer for this suggestion. We re-written this sentence according to the Reviewer’s suggestion. A 497 bp fragment from the 5′ proximal coding region of HvLRR_8-1 was cloned into BSMV: γ to silence HvLRR_8-1 via BSMV-mediated VIGS. (line 431-433, red word)
- L348. It seems the sentience “Constructed a BSMV: HvLRR_8-1 silencing vector, which was inoculated into the leaves of barley seedlings.” is not a complete sentence.
[Response 9] We thank the reviewer for this suggestion. We re-written this sentence according to the Reviewer’s suggestion. The revised text reads as follows: The successfully constructed BSMV: HvLRR_8-1 silencing vector was transfected into barley seedling leaves through the leaf-rubbing inoculation method. (line 435-436, red word).
- L367. The result of Figure 6 D is not mentioned in Result section. Meanwhile, SPAD is not defined.
[Response 10] Thank you for pointing this out. We feel sorry for our carelessness. Based on your comments, the revised text reads as follows: Simultaneously, the relative chlorophyll content at Pg infection sites of the corresponding treatments was measured with a SPAD chlorophyll meter (TYS-4N; Zhongke Weihe Technology Development Co., Ltd., Beijing, China). (line 804-808, red word). Meanwhile, in result L445, the red words indicate that the sentence statements are derived from Figure 6D.

Reviewer 2 Report
Comments and Suggestions for Authors
The manuscript ID: plants-3756417 entitled ”Transcriptome Analysis and Functional Characterization of the HvLRR_8-1 Gene Involved in Barley Resistance to Pyrenophora graminea’’ by Yang et al. performed a comparative transcriptome analysis at different time points using resistant and susceptible barley materials exhibiting differential responses to Pg infection. The authors identified and cloned a leucine-rich repeat receptor-like kinase (LRR-120 RLK) gene HvLRR_8-1 from barley. Moroever, they employed a virus-induced gene silencing (VIGS) system to evaluate its role in Pg resistance. I consider the idea and intention of the article is original with an important scientific result. Minor points need to be considered before acceptance to publication. Below you can find to be considered the most relevant:
Minor points
- The authors need to explain the choice of the Barley HvUBIQUITIN as internal control.
- The authors need to explain the choice of the HvLRR_8-1 for functional analysis
- The paper has some grammar and punctuation errors mainly ‘’space’’ between words and references in the text of manuscript (Lines 80; 83; 105; 109;….).
- The legend of figures should be written not in bold.
- For the bioinformatic analysis, please added the date (day, month and year) of accessing to the different programs.
- Revise the list of references according to the Journal Instructions.
Author Response
Dear Editors and Reviewers:
Thank you for your letter and for the reviewers' comments concerning our manuscript entitled “Transcriptome analysis and functional characterization of the HvLRR_8-1 gene involved in barley resistance to Pyrenophora graminea” (lD: plants-3756417). Those comments are all valuable and very helpful for revising and improving our paper, as well as the important guiding significance to our research. We have studied comments carefully and have made corrections which we hope to meet with approval. Revised portions are marked in red in the manuscript. The main corrections in the manuscript and the responds to the reviewer's comments are as flowing:
Responds to the reviewer's comments:
Reviewer #2:
The manuscript ID: plants-3756417 entitled “Transcriptome Analysis and Functional Characterization of the HvLRR_8-1 Gene Involved in Barley Resistance to Pyrenophora graminea” by Yang et al. performed a comparative transcriptome analysis at different time points using resistant and susceptible barley materials exhibiting differential responses to Pg infection. The authors identified and cloned a leucine-rich repeat receptor-like kinase (LRR-RLK) gene HvLRR_8-1 from barley. Moroever, they employed a virus-induced gene silencing (VIGS) system to evaluate its role in Pg resistance. I consider the idea and intention of the article is original with an important scientific result. Minor points need to be considered before acceptance to publication. Below you can find to be considered the most relevant:
Minor points
- The authors need to explain the choice of the Barley HvUBIQUITIN as internal control.
[Response 1] Thank you for pointing this out. We feel sorry for our carelessness. Based on your comments. Regarding the use of HvUBIQUITIN as an internal reference for validating the reliability of transcriptome results by qRT-PCR in this study, our selection was based on the literature by Yang Ping et al. That study employed BSR-seq (Bulked Segregant RNA-seq) technology to identify genes associated with resistance to barley yellow mosaic disease. The results revealed that HvPDIL5-1 acts as a susceptibility factor for Bymovirus; its presence promotes viral infection, while its loss of function confers resistance to the virus. Notably, the same study also used HvUBIQUITIN as the internal reference gene for barley. We have cited this literature as reference [84] (line 699). Additionally, in our study, we concurrently performed experiments using barley Actin as an internal control. The results demonstrated that HvUBIQUITIN exhibited greater stability, higher consistency across biological replicates, and overall better performance compared to Actin.
- The authors need to explain the choice of the HvLRR_8-1 for functional analysis.
[Response 2] We thank the reviewer for this suggestion. Regarding the rationale for selecting the HvLRR_8-1 gene (HORVU2Hr1G118010) for cloning and functional characterization in this study, our primary objective was to identify novel genes associated with barley resistance to Pyrenophora graminea (Pg). This study employed an integrated strategy combining differential transcriptome analysis and pathway enrichment to screen candidate genes. In the immune barley cultivar Ganpi2, we identified 457 differentially expressed genes (DEGs) potentially involved in plant disease resistance. Further analysis based on KEGG enrichment revealed 11 significantly enriched pathways, from which we prioritized 27 key genes functionally linked to disease resistance and plant-pathogen interactions. Among these, eight genes encoded receptor-like kinases (RLKs). Notably, HvLRR_8-1 (HORVU2Hr1G118010) exhibited the most pronounced upregulation upon Pg infection, with expression peaking at 18 days post-inoculation. This gene also showed the highest fold-change in the resistant cultivar Ganpi2 compared to susceptible controls. Bioinformatic analysis indicated that HvLRR_8-1 encodes an RLK protein harboring a canonical LRR S_TKc domain, which has been reported in Arabidopsis and rice to participate in PAMP-triggered immunity (PTI). However, its functional role in barley’s defense against Pg remains unexplored. Therefore, we selected HvLRR_8-1 as the target gene for cloning and functional validation, aiming to elucidate its mechanistic contribution to barley-Pg immune interactions. Based on your comments, the revised text reads as follows: Integrated analyses of transcription profiling, functional annotation, and disease resistance relevance revealed that receptor-like kinases (RLKs) play pivotal roles in plant disease resistance. It is essential to identify and functionally characterize the corresponding genes from the resistance to Pg in Ganpi2. The HvLRR_8-1 (HORVU2Hr1G118010) gene annotated as receptor-like kinase was selected for next functional analysis. (line 346-351, red word).
- The paper has some grammar and punctuation errors mainly ‘’space’’ between words and references in the text of manuscript (Lines 80; 83; 105; 109;….).
[Response 3] Thank you for pointing this out. We feel sorry for our carelessness. Based on your comments, the revised text reads as follows: line 108-109. Subsequently, these activated proteins interact with receptor-like kinases (RLKs) and other receptors to form functional kinase complexes [36].
L111-113. Among these, leucine-rich repeat receptor-like kinases (LRR-RLKs) represent a central component of plant pattern recognition receptors (PRRs) [37,38]
L133-135. this groundbreaking work has not only established a novel approach for functional gene studies in barley but also laid the foundation for the application of virus-induced gene silencing (VIGS) technology in other monocotyledonous plants [47].
L143. and stem rust resistance genes (TaRLK1/2) [50] in wheat as well as powdery mildew resistance gene (Hsp90) in barley [51].
Line 111-112. Among these, leucine-rich repeat receptor-like kinases (LRR-RLKs) represent a central component of plant pattern recognition receptors (PRRs)
line372. Using the ESPript 3.0 [56] software multiple alignments of the amino acid sequences were performed with the LRR-like proteins from different plant species
- The legend of figures should be written not in bold.
[Response 4] It is really true as Reviewer mentioned that the legend of figures should be written not in bold. Thank you for pointing this out. We have removed all the bolder parts in the annotations of the legend of figures. The main changes that involved bolding the captions on Figure 4E (line 402), Figure 5 (line 425) and Figure 6 (line 457).
- For the bioinformatic analysis, please added the date (day, month and year) of accessing to the different programs.
[Response 5] We thank the reviewer for this suggestion. We feel sorry for our carelessness. Based on your comments, the revised text reads as follows: The physicochemical properties of the protein were analyzed by Expasy Protparam (http://web.expasy.org/protparam/, accessed on 18 June 2025). Transmembrane structural domain prediction for this protein used an online software TMHMM 2.0 (https://services.healthtech.dtu.dk/services/TMHMM-2.0/, online software accessed on 24 May 2025). The secondary structure and tertiary structure of HvLRR_8-1 protein were predicted used the SOPMA program (https://npsa.lyon.inserm.fr/cgi-bin/secpred_sopma.pl, accessed on 18 June 2025) and the SWISS-MODEL (https://swissmodel.expasy.org/, accessed on 16 July 2025). Annotation of HvLRR_8-1 protein structural domains using the SMART (http://smart.embl-heidelberg.de/, accessed on 18 June 2025) protein structural domain database platform. (line 724-734, red word).
- Revise the list of references according to the Journal Instructions.
[Response 6] We thank the reviewer for this suggestion. The reference questions have been modified according to the journal instructions regarding both format and content. (line 869-line1070, red words)

Round 2
Reviewer 1 Report
Comments and Suggestions for Authors
Nice work of the revision